# Rational Use of Protein Supplements in the Elderly—Relevance of Gastrointestinal Mechanisms

**DOI:** 10.3390/nu13041227

**Published:** 2021-04-08

**Authors:** Ian Chapman, Avneet Oberoi, Caroline Giezenaar, Stijn Soenen

**Affiliations:** 1Adelaide Medical School and Centre of Research Excellence (C.R.E.) in Translating Nutritional Science to Good Health, The University of Adelaide, Royal Adelaide Hospital, Adelaide, SA 5000, Australia; ian.chapman@adelaide.edu.au (I.C.); avneet.oberoi@adelaide.edu.au (A.O.); 2Riddett Institute, Massey University, Palmerston North 9430, New Zealand; c.giezenaar@massey.ac.nz; 3Faculty of Health Sciences and Medicine, Bond University, Robina, QLD 4226, Australia

**Keywords:** aging, protein, whey, anorexia, appetite, supplements, sarcopenia

## Abstract

Protein supplements are increasingly used by older people to maintain nutrition and prevent or treat loss of muscle function. Daily protein requirements in older people are in the range of 1.2 gm/kg/day or higher. Many older adults do not consume this much protein and are likely to benefit from higher consumption. Protein supplements are probably best taken twice daily, if possible soon after exercise, in doses that achieve protein intakes of 30 gm or more per episode. It is probably not important to give these supplements between meals, as we have shown no suppressive effects of 30 gm whey drinks, and little if any suppression of 70 gm given to older subjects at varying time intervals from meals. Many gastrointestinal mechanisms controlling food intake change with age, but their contributions to changes in responses to protein are not yet well understood. There may be benefits in giving the supplement with rather than between meals, to achieve protein intakes above the effective anabolic threshold with lower supplement doses, and have favourable effects on food-induced blood glucose increases in older people with, or at risk of developing, type 2 diabetes mellitus; combined protein and glucose drinks lower blood glucose compared with glucose alone in older people.

## 1. Introduction

This review will focus on appetite, feeding, and gastrointestinal responses to protein ingestion in older people. As a background, the high rates of under-nutrition, sarcopenia, and likely sub-optimal dietary protein intake in older people will be outlined, in support of the logical use of protein or protein-rich supplements by older people. The type, amount, and timing of such supplements, as well as possible side effects of these supplements, will be covered. The results of studies with whey protein by our group will be used to illustrate a number of gastrointestinal and cardiovascular responses to protein ingestion and to support some preliminary recommendations about beneficial use of protein supplements by older people. A number of issues remain unresolved.

### 1.1. Aging, Weight Loss, and Undernutrition

Healthy aging is associated with a physiological reduction in appetite and food intake, the so-called “anorexia of aging” [1]; people over 80 years consume about 30% less energy per day than those in their 20s [2]. Consequently, after about 65 years in developed countries, body weight tends to decrease [2]. This age-related weight loss, particularly when substantial and involuntary, has been associated with increased mortality [2]. Adverse factors prevalent in older people can be superimposed on the age-related changes to cause pathological under-nutrition, with even greater associated increases in morbidity and mortality [3]. Rates of under-nutrition increase with age and loss of independence; it is present in up to 45% of community dwelling elderly people, 50–80% in hospital, and 80–100% in residential facilities [3].

The major markers of existing or pending under-nutrition in older people are low body weight (particularly body mass index < 22 kg/m^2^); loss of weight, particularly when involuntary and >5%; substantial loss of muscle mass, which can lead to sarcopenia (see below); reduced muscle strength and function; reduced appetite and food intake; and frailty [2,3].

### 1.2. Aging and Sarcopenia

The weight lost with increased aging is disproportionately made up of lean tissue, particularly skeletal muscle, but also bone mass. In contrast, fat mass increases with ageing, tending to mask the extent of lean tissue loss [2]. After approximately age 30 years, about 5% of lean muscle mass is lost per decade [2]. The loss of lean tissue, when large, can result in sarcopenia, an excessive and damaging loss of muscle. Sarcopenia is present in up to 15% of community-dwelling people over 85 years [4]. Sarcopenia has functional adverse consequences, with increased morbidity due to falls, fractures, infections and other conditions, as well as increased mortality [5].

## 2. Prevention and Management of Under-Nutrition and Sarcopenia in Older People 

A greater appreciation of the high rates and significant adverse effects of undernutrition and sarcopenia, which often co-exist in older people, has led to attempts to prevent and treat these conditions. Both exercise and nutritional measures have been shown to have benefits, particularly when combined [6,7].

### 2.1. Nutritional Measures

Nutritional measures often start with encouragement and assistance to consume greater quantities of usual foods to maintain body weight, the so-called “Food First” approach [8]. These measures may include providing meals at home; supervising food intake and/or having the person eat with others in aged care settings (to counteract reduced appetite and reduced spontaneous food intake); increasing the nutrient and energy density of the food; adding flavour boosters; and fortifying the food with additional fats, protein, and carbohydrates [8].

### 2.2. Nutritional Supplements

While these “Food First” approaches may be sufficient, they may be difficult to implement because of the cost of foods or time/staffing/convenience constraints, and not always successful. Consequently, nutritional supplements are increasingly recommended for, and used by, older people, both as a means of increasing total energy intake (when harmful weight loss is a concern) and to increase protein intake. Mixed macronutrient supplements are used for the former, while supplements of pure protein or very high in protein are used more specifically for the latter.

The use of such supplements offers the opportunity to more closely tailor the timing and composition of added protein intake to what is likely to be most beneficial. It is often recommended that older people take nutritional supplement drinks between and well separated from regular meals so as not to reduce energy intake at those meals; maintenance of weight or even weight gain is a desirable outcome in the many older people with, or at risk of, under-nutrition.

Consequently, specific nutritional supplements, usually commercial preparations, are used widely by older people. When the aim is to maintain body weight and general nutrition, rather than specifically enhance skeletal muscle and function, these usually take the form of a mixture of macronutrients in a drink containing 1–1.5 kcal/mL of protein, fat, and carbohydrates. Frequently used preparations contain about 9–15 gm protein (15–17% of total energy) in a 200–240 mL serving.

### 2.3. Protein Nutritional Supplements

When the aim is to preferentially prevent and/or reverse aging-associated muscle loss and sarcopenia, the use of such mixed macronutrient nutrient supplements may not be sufficient and pure protein or high protein supplements may be used.

## 3. Dietary Protein Requirements in Older People

Recommended dietary protein intakes for adults of all ages, based on nitrogen balance studies, are 0.8 gm/kg body weight in most countries [9], with no adjustments for age or gender. There have been increasing calls, however, for the minimum recommended daily protein intake in older people to be higher than that [10,11]. One reason is concern that the traditional nitrogen balance study methods used to determine appropriate protein requirements in adults of any age underestimate true requirements, with requirements as determined by the newer indicator amino acid oxidation (IAAO) methods up to 25–75% higher [9]. The minimal protein intake for healthy younger adults determined by IAAO is approximately 1.0–1.2 gm/kg/day [9].

Secondly, older people are likely to need higher dietary protein intakes than younger adults for the maintenance of good health, for a number of reasons. These include an age-related reduction in muscle anabolic response to ingested protein; while digestion and absorption of dietary protein is not apparently affected by aging, there is evidence for an age-related reduction in the anabolic muscle response to ingested protein—anabolic resistance (see [10]). This is due to both a redistribution of ingested proteins away from the muscle to splanchnic tissues and a reduced anabolic effect on muscle of the amino acids that do reach the muscle. In addition, catabolic conditions associated with increased muscle breakdown and needs for dietary protein intake become more prevalent with increasing age. These include chronic diseases such as obstructive airways disease, heart failure, renal failure, malignancies, inflammatory forms of arthritis and polymyalgia rheumatica, and acute conditions such as infections and cardiovascular/cerebrovascular events.

Although not conclusive, there is some evidence from balance studies that older people do have higher dietary protein requirements than young adults. One meta-analysis was reported to show approximately 6% higher protein intakes needed to maintain nitrogen balance in people 60 year and older compared with those younger [12], and another approximately 26% higher intakes needed in those over 55 years compared with those younger [13], although neither difference achieved statistical significance due to small subject numbers. Using the indicator amino acid oxidation method mentioned above to study small numbers (12 or less per group) of subjects, protein requirements have been reported as 1.24 gm/kg/day in men over 65 years [14] and 1.15–1.29 gm/kg/day in women over 65 years [15].

### 3.1. Do Men and Women Have Different Dietary Protein Needs?

Women, on average, consume less protein than men, in absolute terms owing to lower body weight, and possibly also less compared with body weight. Women over 70 years in the 2003–4 U.S. NHANES study consumed about 10% less protein per kg body weight than men of the same age, with 50% of these older women reporting daily protein intakes ≤0.9 gm/kg ideal body weight/day, compared with only 25% of older men [16]. These reduced intakes in women are probably in line with reduced needs. The results of several meta-analyses of nitrogen balance studies indicate that adult women need approximately 10% less dietary protein per kg body weight than men to maintain nitrogen balance [12,13], probably because of lower muscle mass relative to body weight. Nevertheless, there were few older subjects in the studies examined in those meta-analyses, and not all studies show lower requirements in women [17]. While protein needs/kg body weight of older women may be lower than those of older men, the difference does not appear to be great. In the interests of simplicity and the absence of evidence that higher protein intakes, if achievable, do much harm (see below), it seems reasonable to maintain gender-neutral protein intake recommendations for older people at this time.

### 3.2. Summary Recommendations for Dietary Protein Intake in Older People

Total Daily Protein Intake

For reasons outlined above, recent recommendations for total daily protein intake in healthy older people are usually in the range of 1–1.5 gm/kg per day. The European PROT-AGE study group, for example, has recommended 1.0–1.2 gm/kg dietary protein for healthy older adults [11]. Others have suggested that even higher intakes (>1.2 gm/kg/day) are needed for maintenance of muscle mass and function [10]. Even higher intakes than these are likely to be needed at times of catabolic stress owing to acute or chronic illness, with the PROT-AGE group recommending 1.2–1.5 g/kg/day for those with acute or chronic illness, and up to 2.0 gm/kg/day for those with severe illness or injury or with marked malnutrition [11].

### 3.3. How Many Older People Need Supplements to Reach the Recommended Protein Intake?

It is unclear how many older adults consume less than the recommended protein intake (either current RDA Recommended Dietary Allowance or the newer recommendations), but probably a substantial proportion. The U.S. NHANES study of 2003–4 reported dietary protein intake substantially lower in healthy older than young adults; in people over 70 years, mean total protein intake was 64.7 gm with a mean intake of 1.0 gm/kg ideal body weight, 27% and 23% respectively, below reported intakes in adults 19–30 years [16]. Over 75% of people over 70 years reported intakes of less than 1.2 gm/kg ideal body weight [16]. Similarly, in the Quebec NuAge study, the mean protein intake of older subjects was very close to 1 gm/kg/day [17].

### 3.4. Possible Adverse Effects of Protein Supplements

Some concerns have been raised about increasing the dietary protein intakes of older people over current recommended levels. These include the following.

#### 3.4.1. Renal Effects

Renal function declines with increasing age [18]. High protein intakes can increase renal filtration and accelerate the progression of established renal disease; restricted protein intakes of 0.3–0.8 gm/kg/day have been shown to delay the progression of established renal disease and are often prescribed for this indication. The adverse effects of high dietary protein intakes on impaired renal function appear to be increased in those with diabetes mellitus, hypertension, or obesity. It is possible, therefore, that increasing the dietary protein intake of older people, whose renal function has already undergone age-related declines, to levels of 1.2 gm/kg/day or higher may induce renal impairment or accelerate the progression of pre-existing renal disease. While this is possible, it is not established that older people with relatively good renal function and without significant risk factors are at such risk. Evidence does not support a relationship between increased dietary protein content and a decline in renal function [19], and high protein diets undertaken for up to 2 years have not been shown to impair renal function in otherwise healthy people [20]. Advanced age alone does not, therefore, appear to be a reason to avoid increased dietary protein intake when indicated, although caution should be exercised in those with pre-existing renal impairment or risk factors for renal function deterioration.

#### 3.4.2. Bone Effects

Previous suggestions that increased dietary protein intake may have adverse effects on bone health by increasing bone mineral loss and increasing the risk of fractures have not been supported by more recent studies [9,21], which have, if anything, shown beneficial effects of dietary protein on bone health.

#### 3.4.3. Post-Prandial Hypotension

Ingestion of nutrients leads to redirection of blood flow from other organs to the splanchnic circulation to aid digestion. As a result, blood pressure (BP) can drop. This is largely prevented in young, healthy adults by an increase in heart rate and other compensatory mechanisms, but in older people, these are less effective. As a result, older people have greater food-induced BP decreases than younger adults [22]. In some, this can be excessive and lead to symptoms of dizziness and, in some cases, falls and cardiovascular events [23]. The excessive drop in BP after food ingestion has been termed post-prandial hypotension and defined as a decrease in systolic BP of 20 mm Hg or more within 2 h of food ingestion [24]. Ingestion of all three macronutrients, alone or together, causes post-prandial BP falls in older people. Available evidence indicates that carbohydrate and protein have equivalent BP lowering effects, although the BP decreases occur sooner after carbohydrate than protein and fat [25]. We have recently reported substantial systolic BP decreases after ingestion of a 70 gm whey protein drink by healthy, older men, with 58% having a decrease of 20 mm of Hg systolic or more within three hours of protein ingestion, with maximum decreases occurring between two and three hours after the drink [26] In contrast, maximum decreases after carbohydrate alone, or mixed drinks, appear to occur earlier. Seventy grams of whey is a high dose, higher than is generally recommended or likely to be ingested by older people (see below), and the hypotensive effects of protein and other macronutrients do seem to be at least partly dose-responsive [27]. Nevertheless, high energy nutrient supplement drinks decrease BP in older people and the hypotensive effects of pure or high protein drinks may be quite prolonged and possibly greater in those already on antihypertensive medications. Appropriate advice and precautions are indicated after such drinks, particularly in at-risk individuals.

### 3.5. Evidence for Benefits of Nutritional Supplements in Older People

A detailed review of this matter is beyond the scope of this review (see [4]). Study methods and results are variable. Not all studies show benefits [17]. Nevertheless, in our view, a number of conclusions can be drawn; the use of such supplements is generally safe (see above) and relatively easy to implement. Few, if any, adverse effects of such supplements have been reported; the use of these supplements by older people improves nutritional intakes and is associated with weight gain and an increase in lean body mass in many cases and, in some cases, improvements in muscle function, hospitalisation rates, and even death rates [28]. The greatest benefits of taking nutritional supplements are obtained by the most undernourished older people [29].

For these reasons, and because many older people appear to have a suboptimal dietary protein intake, protein or protein-rich supplements are increasingly recommended to older people. 

In summary, there is evidence of the following:Older people have higher dietary protein needs than young adults, particularly if they have lost weight and/or are undernourished, are sarcopenic, or have acute or chronic medical conditions that contribute to muscle catabolism. These requirements are in the range of 1.2 gm/kg/day or higher.Many (probably the majority of those >70 years) older people do not consume these minimum protein requirements. It can be difficult for older people to increase their dietary protein intake by increasing their intake of usual foods owing to anorexia and the cost of high protein foods.Protein supplements increase muscle mass and strength, and may also reduce morbidity and mortality, particularly in undernourished and/or sarcopenic older people.

### 3.6. Does the Type of Protein or Amino Acid in the Supplement Matter?

Yes, it probably does. Not all ingested proteins are the same, particularly in terms of their anabolic effects on skeletal muscle (see [9] for review). The available evidence indicates that branched chain amino acids, particularly leucine, are the most effective amino acids in stimulating muscle protein synthesis (MPS). Consistent with this, in one longitudinal study, older people (>65 years) with dietary leucine intakes in the upper quartile had preservation of lean body mass over 6 years, whereas those with intakes in the lowest quartile had loss of lean body mass [30]. The extent of MPS is proportional to the peak leucine plasma concentration after protein ingestion [31].

Animal-derived dietary proteins appear to be stronger in stimulating MPS than plant-based proteins, and milk-based proteins in particular are a good source of leucine containing proteins [9]. Whey protein, obtained from milk in the cheese making process, is high in leucine, about 10–15%, and more rapidly absorbed than casein, factors that may contribute to its greater stimulatory effect than casein on MPS in older men [32]. The maximum stimulatory effects on MPS probably occur after ingestion of 2–2.5 gm leucine in young and middle aged adults, with little additional effect of higher intakes. Ingestion of 20 gm whey protein concentrate at one time is probably sufficient to optimise MPS in young and middle aged adults [33]. In contrast, because of anabolic resistance, older adults appear to need 30–35 gm or more of whey to achieve similar anabolic effects [34].

The effects of protein supplements to increase muscle mass and strength and to have functional benefits appear greatest in those most at risk; i.e., malnourished, sarcopenic, or at risk of sarcopenia. Several interventional studies have demonstrated that combined dietary supplements of whey protein and leucine increase muscle mass and strength and improve function in older sarcopenic adults; see [9]. A recent meta-analysis reported increased lean body mass without an increase in strength in sarcopenic older adults taking 2–7.8 gm of leucine supplements/day [35]. It is difficult to separate the effects of leucine supplements from those of other amino acids or proteins such as whey when they are given in combination. It may be, particularly in the case of already leucine-rich whey protein supplements, that they have benefits in addition to those of leucine, and that, when a sufficient amount is ingested, further leucine fortification has little further benefit.

### 3.7. How Much Protein Should There Be in the Supplement?

Owing to age-induced “anabolic resistance”, older people probably require 30–45 gm protein per serving to stimulate muscle protein synthesis after that meal, whereas lower doses (≤20 gm) are sufficient in young adults (see above and [36]). The consumption of 1–2 daily meals with protein content of 30 to 45 g may be an important strategy for increasing and/or maintaining lean body mass and muscle strength with aging [36]. There is evidence that older men and women with more-evenly mealtime distributed protein intakes have higher muscle strength irrespective of their total protein intake [17]. It seems reasonable, therefore, that if protein supplements are being used for their effects on muscle, they be used in a way that provides at least 1.2 gm/kg/protein per day (food plus supplement) with at least two protein intake episodes/day (food plus supplement) containing at least 30 gm protein each. As the post protein BP drop, which could be potentially harmful, appears to be dose-responsive, it is also likely that smaller, twice daily or even more frequent, doses of protein supplement will have more beneficial effects on BP than once daily larger doses.

### 3.8. Timing of Protein Intake Relative to Exercise

Both protein ingestion and resistance exercise independently stimulate muscle protein synthesis in older people, although the response to both is blunted in the elderly [37,38]. These two stimuli have synergistic anabolic effects on MPS, particularly when the protein is ingested soon after the resistance exercise [39]. While ingestion of approximately 20 gm of a high quality protein is sufficient to maximize skeletal muscle protein synthesis rates during recovery from resistance-type exercise in younger adults, doses up to 40 gm or possibly even higher are needed in older adults [37,38].

## 4. Gastrointestinal Responses to Protein Ingestion: Effects of Aging

Appetite and food intake in free living humans are dependent on a complex interplay of environmental factors and central and peripheral physical mechanisms. The latter mechanisms include intra-gastric and small intestinal sensory and motor functions and their interactions [2]. Their study has been a focus of our group. We have used whey protein drinks and intra-duodenal infusions of whey protein to investigate peripheral responses to protein in young and older adults and have identified a number of age-related differences between these age groups in gastrointestinal responses to protein ingestion. These studies have involved older people predominantly of Anglo-Saxon background. The doses of oral whey used have mainly been 30 gm and/or 70 gm, which is of significance as 30 gm appears to approximate the amount required in older people to stimulate muscle protein synthesis (see above), while higher doses are used by some older people.

Older people have reduced appetite compared with young adults and consume less food. There is no change, however, with aging in the preference for particular macronutrients; i.e., the percentage of total energy ingested as protein does not seem to change with increasing age [40].

### 4.1. Effect of Aging on Appetite and Feeding Responses to Whey Protein

There is evidence that protein is the most satiating of the macronutrients in young adults [41], although this effect may be less in women than men [42]. Our studies have demonstrated that healthy aging is associated with a marked and significant reduction in the satiating effects of whey protein, administered by both intra-duodenal [43] and oral routes [44,45].

#### 4.1.1. Intra-Duodenal Whey

Healthy aging is associated with a marked reduction in the suppressive effect of whey protein administered directly into the duodenum, on appetite and subsequent food intake. Sixty minute intra-duodenal infusions of 8 gm (0.5 kcal/min), 24 gm (1.5 kcal/min), and 48 gm (3 kcal/min) of whey had a dose-responsive suppressive effect on subsequent ad libitum food intake in young men, whereas older men experienced suppression of food intake only after the 48 gm infusion (~33% vs. 17% suppression by 48 gm whey in young vs. older, *p* < 0.05) [43]. Baseline hunger ratings were lower in the older than young men and were suppressed less by the protein infusions in the older than young men, consistent with the reduced effects of intra-duodenal fat and carbohydrate infusions on appetite in older people [46].

#### 4.1.2. Oral Whey

We have reported that healthy, non-obese, young men, but not women, experience significant suppression of hunger ratings and ad libitum food intake three hours after 30 gm and 70 gm whey protein drinks [42,44,47]. In contrast, and consistent with their reduced responses to intra-duodenal whey, healthy, non-obese men and women over 65 years experience little reduction in hunger and no suppression of ad libitum food intake three hours after either 30 gm or 70 gm whey drinks [44,47,48,49,50].

This age-related reduction in the satiating effects of whey drinks is observed in our studies largely irrespective of the timing of the whey drink relative to later ad libitum food intake at test meals. Thirty gram whey protein drinks do not suppress appetite ratings or subsequent ad libitum food intake in healthy older people immediately after the drink, or 35 min, 1 h, 2 h, 3 h, 265 min, and 510 min after the drink [44,45,49,50,51]. In only one study have we detected any reduction in subsequent food intake by older people after a whey drink [45]. In that study, older men received a 30 gm or 70 gm whey drink and then ate adlibitum at breakfast (30 min later), lunch (265 min later), and dinner (510 min later). There was no reduction in food intake compared with the control day at any of the three meals on the 30 gm day. On the 70 gm day, there was no effect on breakfast intake, but a 15% reduction in lunch energy intake (*p* < 0.05 vs. control), followed by a compensatory 7% increase at dinner.

When intake from whey drinks plus subsequent food intake is calculated, absent (usually) or only minor suppression of subsequent food intake by whey drinks has resulted in consistent findings of increases in total energy intake and even greater proportional increases in protein intake after whey drinks in our short-term studies of older people. For example, in the study above, where whey drinks were taken before breakfast and men were studied for the rest of the day [45], there were non-significant 4% and 3% increases in total energy intake on the 30 gm and 70 gm days, respectively, compared with the control day. Total daily protein intake was increased significantly by the whey drinks in a dose-responsive manner, with increases almost equal to the protein content of the whey drinks (+31 gm on the 30 gm whey drink day, +62 gm on the 70 gm whey drink day, *p* < 0.001 vs. control [45]).

These findings suggest that it should be possible to give enough extra protein to older people to preserve or increase muscle mass and function without suppressing energy intake and promoting weight loss, particularly if they are encouraged to continue their usual non-supplement food (energy) intake.

### 4.2. What Is the Best Timing of the Protein Supplement Use by Older People?

Because older men can increase their protein intake in a single episode into the range of 30–40 gm, enough to maximize the protein’s anabolic effects on muscle without the timing of that supplement’s ingestion making much if any difference to subsequent appetite and food intake, these supplements can probably be ingested as a between-meal supplement, close to or even with meals. The effects in women are likely to be similar [50], but require further study. The effects of more frequent protein doses across the day also need to be determined.

Indeed, if the supplement is given with a protein-containing meal, instead of between meals, less supplementary protein is needed to reach the 30–40 gm anabolic threshold described above. Older men in the NuAge study, for example, had a mean daily protein intake of 1 gm/kg (mean of 18 gm at breakfast and 23 gm at lunch and dinner) [17]. Their protein intake could be increased to 1.33 gm/kg/day with a total protein intake of 35 gm per meal two meals a day by adding as little as 12 gm protein with each of lunch and dinner. A pragmatic measure, to allow for those with below average dietary protein intakes, would be to take 20 gm of a protein supplement with the two meals each day already containing the most protein. This dose is unlikely to have many, if any, side effects for most older people.

### 4.3. Effect of Aging on Gastric Function and Emptying

Healthy aging is associated with changes in gastric function. These include reduced perceptions of proximal gastric distension and delayed gastric accommodation [52], probable changes in the intra-gastric distribution of food after its ingestion, greater stimulation of phasic pyloric pressure waves by intra-duodenal lipid [46], and slowing of gastric emptying. It is unclear, however, how much, if at all, these changes contribute to the age-related reduction in the satiating effect of ingested protein and responses to protein supplements in older people. It is possible, however, that they may contribute to age-related reduced appetite and food intake.

#### 4.3.1. Intra-Gastric Food Distribution and Antral Area

After food is ingested, it is distributed throughout the stomach, with varying amounts in the proximal versus distal stomach (antrum). Antral distension appears to be a greater determinant of satiety and satiation than proximal gastric distension; the more distal the intra-gastric distribution of food, the greater the antral distension, the greater the sensations of fullness, the less the hunger, and the lower the subsequent food intake [53]. Owing to impaired receptive relaxation of the gastric fundus [52] and other factors, food is distributed more distally in older people, as indicated by them having larger antral areas than young adults after ingestion of the same mixed macro-nutrient loads [53]. As a result, they probably experience greater fullness and lower hunger ratings. While the more distal movement of food after its ingestion thus probably contributes to reduced hunger and increased fullness in older people, it is unclear how these changes could contribute to the reduced suppression of appetite ratings and food intake after oral ingestion of protein, alone or combined with other nutrients.

#### 4.3.2. Gastric Emptying

The oral ingestion of nutrients, irrespective of their type, slows gastric emptying. Gastric emptying of non-nutrient drinks is slightly slower in older than in young adults [44,45,46,47,48,49,50,51,52,53,54]. Whey protein drinks in doses of 30–70 gm slow gastric emptying in a dose-responsive manner [42,44,50], and probably slow it more in older than in young adults; the stomachs of healthy older men emptied whey into the duodenum at approximately 0.8 kcal/min compared with 1.0 kcal for young adult men (*p* < 0.05) in one study [44]. Gastric emptying of whey protein drinks is faster in young, non-obese men than women [42], but this sex difference is no longer present in healthy people over 65 years [50]. Gastric emptying of whey drinks occurs at a similar rate in obese young and older men [49], with neither age group having suppression of appetite or food intake 3 hours after ingestion of 30 gm whey—a suppressive dose in young, non-obese men [42,44]. The findings suggest that obesity may blunt the effects of whey on both food intake and the slowing of gastric emptying.

Delayed gastric emptying results in more food in the stomach at any given time after food ingestion than would otherwise be the case. This increases gastric distension and, depending on the distribution of the food within the stomach, might be expected to produce greater feelings of fullness and reduced subsequent food intake. Consistent with this, the intentional creation of gastric distension and fullness by implanting gastric balloons has had some success in the treatment of obesity [55]. Greater gastric distension after eating, due to slower gastric emptying, in older than in young adults, might thus be expected to cause greater post-eating suppression of appetite and food intake in older than in young adults. Existing evidence does not suggest, however, that this is so. Perceptions of gastric distension are less in older than young adults [52], and healthy older people have fullness ratings (unlike hunger ratings) 3 h after a whey drink, at the start of an ad libitum meal, that do not relate significantly to energy intake at that meal [44]. Furthermore, as outlined above, older people have less, not more, suppression of appetite and food intake by whey drinks than younger adults, despite slower gastric emptying and presumably greater gastric distension at any given time after the drinks. Our groups’ studies have largely involved administering the ad libitum test meal 3 hours after the whey test drink, in order to allow the full effect of gastric mechanisms to be studied. The stomach is empty or almost empty by then after both 30 gm and 70 gm whey protein drinks [44]. While it is possible that, if the meal had been given earlier, greater suppressive effects on intake due to greater gastric distension would have been present, this is not supported by our finding of a lack of suppression by 30 gm whey of food intake immediately after the drink and hourly up to 3 hours after the drink [51].

#### 4.3.3. Small Intestinal Satiety Mechanisms

As aging has similar qualitative effects on appetite and food intake after intra-duodenal protein (whey) to those of oral whey (see above), and the age-related changes in gastric mechanisms do not appear to explain the reduced suppression of appetite and food intake by protein in older people, it seems likely that the reduced age-related changes in response to protein are mediated mainly by reductions in the satiating effects of the protein after it enters the small intestine. Nevertheless, slower gastric emptying in older people may modify these post-gastric effects by delaying their onset and prolonging their duration. Sixty minute duodenal infusions of 0.5 kcal/min and 1.5 kcal whey/minute do not suppress subsequent food intake in healthy older men, but 3 kcal/min infusions do, albeit less in older than in younger men [43]. As healthy older men empty whey drinks from the stomach at a rate of approximately 0.8 kcal/min [44], it is likely that, even if a very high protein load is taken in drink form, for most older people, their age-related slowing of gastric emptying and markedly reduced effect of whey once it enters the duodenum combine to result in no suppression of appetite or subsequent energy intake. In contrast, duodenal infusions of whey in doses of 0.5, 1.5, and 3 kcal/in have dose-responsive suppressive effects on energy intake in young men, who empty whey drinks from the stomach faster than older men (~1 kcal/min) [44]. Together, these findings explain the suppressive effects of both 30 gm and 70 gm whey drinks on subsequent energy intake in young, but not older adults [44].

### 4.4. Selected Hormones

Ingestion of protein, either orally or by infusion directly into the duodenum, results in changes in circulating concentrations of a number of hormones with definite or possible effects on appetite and subsequent food intake. Among them, concentrations of cholecystokinin (CCK), insulin, glucagon, gastric inhibitory peptide (GIP), glucagon-like peptide-1 (GLP-1), peptide tyrosine-tyrosine (PYY), and amino acids increase, while glucose does not change and ghrelin concentrations decrease [42,47,48,50,56,57].

A number of these small intestinal responses to protein ingestion differ between healthy older and younger adults and these age-related changes in turn possibly affect the responses to protein or protein containing supplements in older people.

#### 4.4.1. Cholecystokinin (CCK)

CCK is released by the small bowel after nutrient ingestion and acts to slow gastric emptying and reduce food intake. Circulating CCK concentrations increase less after oral protein ingestion in young women than in young men, which may be one reason for the lower satiating effect of whey protein in young women than men [42]. Circulating CCK concentrations are higher in healthy fasting older than young adults [58,59], and older adults retain their sensitivity to the satiating effects of exogenous CCK [60]. Increased CCK activity may thus be a cause of the anorexia of ageing and reduced hunger pre-meals observed in older adults. Increases in circulating CCK concentrations are at least as great [53], if not greater [58], after whey drinks in healthy older than young adult men and women. The cause of this possibly greater rise is not known; it may relate to slower transition of whey through the small bowel in older people and, therefore, more prolonged contact with the CCK releasing cells. Ageing-related increases in CCK secretion and action are unlikely, however, to contribute to the reduced suppression of appetite and food intake by whey protein observed in older people compared with young adults. The opposite might be expected. The exact role of CCK in mediating age-related responses to protein supplements remains to be determined.

#### 4.4.2. Glucagon-Like Peptide 1 (GLP-1)

GLP-1 is released by the small bowel and colon in response to food ingestion. Like CCK, it slows gastric emptying and has satiating effects. Fasting GLP-1 concentrations are higher in healthy older than young ageing [58,59], which may thus contribute to lower basal (fasting) hunger in older people. Circulating GLP-1 concentrations appear to increase to a similar extent after whey drinks in young and older adults [54,55,56], which does not support a role for the lesser suppression of food intake by whey protein in older adults.

#### 4.4.3. Gastric Inhibitory Peptide (GIP)

Along with GLP-1, GIP is an incretin that plays roles in the control of glucagon, insulin, and blood glucose concentrations. It is not clear what role GIP plays in the control of appetite and feeding, but it may have some effect to stimulate food intake [61]. Circulating fasting GIP concentrations are not affected by normal aging, but GIP concentrations increase more after oral whey ingestion in older than in young adults [45,46,47,48,49,50,51,52,53,54]. This greater increase might act to reduce the whey-induced suppression of food intake that would otherwise occur. It might also act to limit blood glucose concentration increases in older people after protein is co-ingested with other nutrients.

#### 4.4.4. Insulin and Glucose

Insulin plays a key role in glucose homeostasis, while its role in appetite and feeding control is less clear. Oral ingestion of protein, including whey, stimulates insulin secretion in a dose-responsive manner [58]. While oral ingestion of protein on its own has little, if any, effect on blood glucose concentrations, its co-ingestion with glucose by non-elderly adults with type 2 diabetes results in significantly smaller increases in blood glucose concentrations than ingestion of the same amount of glucose on its own [62]. The stimulation of insulin secretion by whey drinks is not apparently affected by aging, and remains robust [58]. We have recently found that co-ingestion of 30 gm whey protein with 30 gm glucose in drink form significantly reduces the increase in blood glucose concentrations compared with ingestion of 30 gm glucose alone (peak glucose 7.4 vs. 9.0 mmol/L, *p* < 0.01) in men over 65 years [27], and are now extending these studies to older people with type 2 diabetes. These findings suggest that moderately high whey protein intake together with carbohydrate might improve postprandial glycaemia in older people, particularly those with diabetes, and provide an additional benefit of taking a protein supplement with, rather than between, meals.

#### 4.4.5. Glucagon

Circulating concentrations of glucagon are not affected by aging [54,55,56]. Whey protein drinks act to increase glucagon concentrations, to a similar degree in older and young adults [58].

#### 4.4.6. Ghrelin

Ghrelin is an orexigenic hormone secreted by the enteroendocrine cells of the gastrointestinal tract, particularly the stomach. Circulating concentrations are highest in the fasting state and decrease after food ingestion. Fasting circulating concentrations may be slightly lower in older than in young adults, and thus contribute to the anorexia of aging [63,64,65]. Ghrelin concentrations are suppressed to a similar degree after whey drinks in healthy older and young adults [58], so it is not clear what role, if any, ghrelin plays in mediating age-related reductions in feeding responses to protein supplements.

## 5. Future Directions

Future studies should focus on women as well as men to determine whether our findings in older men also apply to women. They should also determine whether the results of our short-term studies with protein supplements, such as post-supplement blood pressure drops and failure to suppress appetite and food intake, are replicated in studies of longer term protein supplement use. Further studies, both short- and longer-term, of the effect of protein when co-ingested with carbohydrates on glucose metabolism in older people with and without diabetes mellitus would also be of interest.

## 6. Conclusions

Nutritional supplements, including pure-protein or protein-enriched drinks, are increasingly used by older people to maintain body weight and nutrition, and specifically to prevent or treat loss of muscle function, sarcopenia, and frailty. Daily protein requirements in older people appear to be higher than those of young adults—in the range of 1.2 gm/kg/day or more. Many older adults do not consume this much protein in their usual diet and are likely to benefit from an increase. Protein supplements appear relatively safe, although care should be taken in those with or at risk of renal impairment, or prone to post-nutrient hypotension. Protein supplements (alone or in a mixed macronutrient supplement) in doses sufficient to reach the above daily intake are likely to be beneficial, perhaps best taken twice daily, if possible soon after resistance exercise, in doses that achieve protein intakes of 30 gm or more per episode. Our study results suggest that it is probably not important to give these supplements between meals, because they have little, if any, suppressive effect on appetite and later food intake in older people, owing to age-related changes in gastrointestinal and other mechanisms that are as yet poorly understood. Adding protein supplements to usual food intake is very unlikely to reduce energy intake and instead is likely to increase overall energy and protein intake, particularly if encouragement is given to continue usual food intake. There may even be benefits in giving the supplement with meals, to achieve protein intakes above the effective anabolic threshold with lower supplement doses, and have favorable effects on food-induced blood glucose increases in older people with, or at risk of, developing type 2 diabetes mellitus.

Further studies are indicated to determine the following:The acute effects of whey and other proteins, when co-ingested with other macronutrients, with and between meals, on appetite and food intake in older people.The longer term effects of protein ingestion, alone and combined with other macronutrients, on appetite, food intake, and glucose homoeostasis, in older people; i.e., whether effects observed in acute studies persist with longer-term administration.

A better understanding of the mechanisms underlying the reduced suppression of appetite and food intake by protein and other macronutrients in older compared with younger adults, whether gastro-intestinal, central, or both, can be used to develop ways of improving nutrition in at-risk older individuals. Although our studies with whey have identified a number of age-related changes in gastro-intestinal responses to protein ingestion, it is not yet clear exactly how they act to mediate age-related changes in appetite and feeding.

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
