# Peer review of "Rational Use of Protein Supplements in the Elderly—Relevance of Gastrointestinal Mechanisms"

_nutrients, 2021, doi:10.3390/nu13041227_

Round 1

Reviewer 1 Report

Reviewer’s Comments:

The manuscript " Rational Use of Protein Supplements in the elderly –relevance 2 of gastrointestinal mechanisms" by Chapman et al presents an interesting review article that highlights the effects of protein supplements’ ingestion on gastrointestinal responses.

  1. Is there any information available about the ethnicities of the study population?
  2. Please add appropriate references to the “Nutritional Supplements” and “Protein Nutritional Supplements” sections.
  3. In a simplified table format, please provide the available protein supplements, their ingredients and benefits in the reviewed study population.
  4. Please add a brief paragraph on “future directions to this study” at the end of the discussion/conclusions section?
  5. Please be consistent with the style of references. For example, in references 10, 13, and 14, the page number style is different.

Author Response

We thank the reviewers for their comments and provide the following responses.

Both reviewers request additional information and detail.  We have provided as much as we can but do not want to lengthen an already substantial paper too much – we note that reviewer 3 has also asked us to shorten the paper. We consider some of the information requested, while of interest, may move the emphasis of the paper away from the main points we wish to make, and what we were asked by the commissioning editors to address.

For example, although the prevention and treatment of sarcopaenia is an important reason why older people may be encouraged to take protein supplements, as mentioned in our paper, details of sarcopaenia, the underlying mechanisms, diagnostic tools etc., are not intended by us to be a major part of this paper. We refer to those matters throughout the early part of the paper, with appropriate references (particularly reference 2 by Soenen et al), which can be accessed by the interested reader.

Reviewer 1.

Re the ethnicities of the populations we have studied. These are largely representative of the older -age population living in our city of Adelaide i.e. largely anglo-saxon. We have now included a sentence mentioning this in the section “Gastrointestinal Responses to Protein Ingestion: Effects of Aging”

Re the request for a table providing details of available supplements. There are multiple such available supplements, probably running into the hundreds for those commercially available alone. Space limitations do not allow us to provide such a comprehensive table.

We have added a brief paragraph at the end on future directions, as requested.

We have corrected the referencing.

Reviewer 2 Report

In this article, I. Chapman and colleagues review the use of protein supplements in the context of ageing, focusing on gastrointestinal mechanisms. The review is complete and very well written.

A minor comment concerns the organization of the different parts. There is a first part (1. Introduction), a third part (3. Conclusion) but the second part is missing and could be organized with better differentiated sub-parts. In addition, adding a synthesis figure would be valuable for the review.

Author Response

We thank the reviewers for their comments and provide the following responses.

Both reviewers request additional information and detail.  We have provided as much as we can but do not want to lengthen an already substantial paper too much – we note that reviewer 3 has also asked us to shorten the paper. We consider some of the information requested, while of interest, may move the emphasis of the paper away from the main points we wish to make, and what we were asked by the commissioning editors to address.

For example, although the prevention and treatment of sarcopaenia is an important reason why older people may be encouraged to take protein supplements, as mentioned in our paper, details of sarcopaenia, the underlying mechanisms, diagnostic tools etc., are not intended by us to be a major part of this paper. We refer to those matters throughout the early part of the paper, with appropriate references (particularly reference 2 by Soenen et al), which can be accessed by the interested reader.

Reviewer 2.

We have renumbered the main parts 1-6 as requested.

A graphical representation of the main points has been attached separately.

Reviewer 3 Report

This is a nice read, but there are some points that should be considered. As to the protein deficiency: What is the real background for the age-dependent  protein/muscle loss?

  • Is this measurable with some biomarker? Myostatin levels in plasma?
  • is this measurable in terms of muscle volume? or function/strength?
  • Is it only based on epidemiological (weak) data?

So basically how do we know that the target intake is 1.2 g is the right daily intake.

We need to understand how we should scientifically attack the problem wirth sarcopenia (similar to osteopenia/osteoporosis). Is there a specific muscle/muscle group that should be representative to measure on CT scan for standardized sarcopenia? Why cannot DXA test be used for lean body mass?

What the world needs is a marker for sarcopenia. 

Line 209 should read 20 mm Hg.

Psychologically, I believe it would be of interest to know what the togetherness with other people for eating effects the intake. You eat with your friends.  Loneliness with aging has major impact on protein intake.

I would recommend shortening of the paper in order to profile a clear message. Not describe what anyone has found. That does not move the frontier. Make clear statements.

Author Response

We thank the reviewers for their comments and provide the following responses.

Both reviewers request additional information and detail.  We have provided as much as we can but do not want to lengthen an already substantial paper too much – we note that reviewer 3 has also asked us to shorten the paper. We consider some of the information requested, while of interest, may move the emphasis of the paper away from the main points we wish to make, and what we were asked by the commissioning editors to address.

For example, although the prevention and treatment of sarcopaenia is an important reason why older people may be encouraged to take protein supplements, as mentioned in our paper, details of sarcopaenia, the underlying mechanisms, diagnostic tools etc., are not intended by us to be a major part of this paper. We refer to those matters throughout the early part of the paper, with appropriate references (particularly reference 2 by Soenen et al), which can be accessed by the interested reader.

Reviewer 3

“So basically how do we know that target intake is 1.2gm …

This is covered in the section “Dietary Protein Requirements in older people”. We present our interpretation of the literature with supporting references. We believe it is sufficient for the purposes of this paper,  and that  our interpretation of the available literature provides a balanced and appropriate interpretation of available data.

Is there a specific muscle group that should be representative to measure on CT scan for standardized sarcopenia? Why cannot DXA test be used for lean body mass?

As requested by the commissioning editors the paper has focused on rational use of protein supplements in the elderly and gastrointestinal mechanisms. Sarcopenia is mentioned and discussed, as part of the background as to why protein supplements are used, but is not the main focus of this paper, and because of space constraints we do not feel able to expand this discussion further. Reference 2 by Soenen et al covers this in more detail, including discussion of the definitions of sarcopenia based on body composition measurements. DXA is already used  for this purpose and is discussed in ref 2 in our paper (sarcopenia exists when skeletal mass is “more than 2 standard deviations below the young adult mean as measured by dual-energy X-ray absorptiometry”). Despite work by our group and others using DXA for this purpose, and by our group to develop ultrasound methods to measure muscle mass of specific groups, we do not think a more detailed discussion of which muscle groups should be measured by which imaging techniques was called for in this paper which focuses on protein supplements and gastrointestinal mechanisms involved in responses to them.

We have corrected 20 mg Hg to 20 mm Hg

The reviewer makes an interesting and important point about psychological factors and eating together. We agree this is important and have mentioned it in the section on nutritional measures.

 The reviewer asks us not to “describe what anyone has found” and “Make clear statements”.

We are not clear what is intended by those requests. We were asked by the commissioning editors to include details of our group’s work, which along with details of work from other groups we consider relevant, we have done. We hope our statements are reasonably clear.
